# PROBING CONFIDENCE REGIONS FOR EARLY EXITS IN CHAIN-OF-THOUGHT

## ABSTRACT

Chain-of-Thought (CoT) has demonstrated remarkable problem-solving capabilities in many large language models (LLMs), but their reasoning processes often exhibit substantial redundancy. To mitigate these issues, various approaches have been explored to improve reasoning efficiency. In this paper, we focus on early exit methods which are lightweight and can be seamlessly adapted to various methods. These methods typically trigger an early exit based on different types of signals, including localized criteria like single-step confidence scores, as well as stabilization of an intermediate trial answer over multiple steps. However, we observe that they often struggle to confirm whether the underlying reasoning process is complete or sound. This paper pioneers to diagnose the reasoning state of CoT through the global dynamics of its entropy. We reveal a consistent pattern: CoT generation begins in a high-entropy uncertainty region before transitioning to a stable, low-entropy confidence region. We demonstrate that the transition into this confidence region strongly correlates with a complete reasoning process. Based on this insight, we propose COnfidence Region Exits (CORE) to stop the CoT when the model enters the confidence region. Experiments demonstrate that our approach achieves a superior trade-off between computational cost and accuracy among early exit methods across various models including Deepseek-R1-Distill-Qwen-7B, Qwen3-4B-Thinking-2507, and Qwen3-14B in AIME24, AIME25, and GPQA datasets [1]. We believe that this method can serve as a strong efficient reasoning method and provide insights for understanding CoT.

## 1 INTRODUCTION

Chain-of-Thought (CoT) reasoning (Wei et al., 2022) has demonstrated remarkable capabilities in Large Language Models (LLMs) by solving a wide range of complex tasks (Shao et al., 2024; Guo et al., 2025; Wang & Chen, 2023; Team et al., 2025; Anil et al., 2023). Instead of directly generating a final answer, CoT externalizes its reasoning process, articulating a series of intermediate, logical steps that lead to a conclusion. This methodological shift has unlocked substantial performance gains (Merrill & Sabharwal, 2024), particularly tasks requiring multi-step deliberation such as math, reasoning, and coding. By guiding the model to think step by step, CoT transforms problems that were previously intractable for LLMs into solvable challenges, marking a pivotal advance in developing more robust and reliable reasoning systems.

However, enhanced reasoning performance comes at the cost of high token usage (Chen et al., 2024). This inefficiency creates two critical problems: It drives up economic costs through expensive API calls and hardware demands (Zellinger & Thomson, 2025), and it degrades the user experience with high latency in interactive applications (Liang & Tong, 2025). This combined overhead is a key barrier to the practical, widespread use of our most powerful LLMs.

Many approaches have been proposed to improve the reasoning efficiency of LLMs (Hao et al., 2024; Zhang et al., 2025b; Shen et al., 2025; Yu et al., 2025; 2024; Tu et al., 2025; Li et al., 2025). This paper focuses on the early exit framework, which is compelling because of its lightweight and seamless adaptability to various models. With this framework, existing methods typically rely on heuristics to dynamically halt generation. Some focus on answer confidence, terminate the process

---

[1]Our code will be public once accepted.

when the model exhibits high confidence (Yang et al., 2025; Huang et al., 2025). Some focus on stable probing answer, terminate when the same answer is repeatedly generated (Fu et al., 2024). While effective in certain scenarios, we argue that neither approach provides a definitive measure of reasoning completion. A high-confidence answer can be prematurely asserted, and a stable answer may simply indicate a reasoning loop rather than a sound conclusion.

In this paper, we introduce a novel framework for early exit Chain-of-Thought (CoT) reasoning grounded in its underlying entropy dynamics. To the best of our knowledge, our work is the first to investigate the evolution of entropy throughout the CoT process. Our key discovery is that the reasoning process is not monolithic but consistently bifurcates into two distinct region: an initial, high-entropy uncertainty region for exploration, followed by a stable, low-entropy confidence region where the reasoning converges. We establish that the transition into this confidence region is a reliable indicator of a complete reasoning chain. Leveraging this insight, we propose a lightweight, training-free halting criterion that terminates generation precisely when the model enters confidence region, effectively identifying the moment reasoning is complete. This principled approach avoids both premature exits during deliberation and redundant verbosity after a solution is found, ensuring efficiency without compromising accuracy.

We conduct comprehensive experiments on three LLMs including DeepSeek-R1-Distill-Qwen-7B, Qwen3-4B-Thinking-2507, and Qwen3-14B across three challenging benchmarks: AIME24, AIME25, and GPQA Diamond. Experiments demonstrate that our method achieves a Pareto-optimal trade-off compared to all other early exit methods. We believe this work can provide a novel perspective to understand the CoT process. This work offers a novel perspective for understanding the internal dynamics of the CoT process and provides a more principled and robust method for efficient reasoning.

## 2 RISK OF PREVIOUS EARLY EXITING METHOD

LLMs can improve their reasoning capabilities by generating intermediate steps through CoT. In this manner, the model generates explicit intermediate reasoning steps before producing a final answer and enables substantial performance gains on complex tasks. For every input $X$, the model auto-regressively generates a sequence of reasoning steps $T_1, T_2, \cdots, T_n$. Each step $T_i$ is a sequence of tokens (e.g, fixed interval of tokens) sampled from the model's predictive distribution, conditioned on the input and all preceding steps:

$$T_i \sim \text{LLM}(T_i|X, T_{1:i-1}),$$

where $T_{1:i-1} = T_1, T_2, \cdots, T_{i-1}$. This process continues until a complete reasoning chain $T_{1:n}$ is formed, which is then used to derive a final answer.

Despite its effectiveness, the generation of the reasoning chain can be computationally costly due to unnecessarily long traces. To address this issue, many early exit methods are proposed. In this section, we formalize the early exit mechanism and introduce potential risks of previous methods.

### 2.1 EARLY EXIT MECHANISM

Current early exit methods for CoT operate within a two-stage framework, which first involves probing the model's intermediate explicit reasoning state for a trial answer, and subsequently applies a stopping criterion to determine generation halt for immediate final answer derivation.

**Intermediate Answer Probing.** This mechanism aims to quantitatively monitor the CoT process by eliciting intermediate answers from intermediate thoughts. Specifically, following each step $T_i$, we prompt LLMs for an intermediate answer $A_i$ by appending an answer-inducing prompt, $P_{\text{INDUCE}}$.

$$A_i \sim \text{LLM}(X, T_1, T_2, \cdots, T_i, P_{\text{INDUCE}}). \tag{1}$$

For example, $P_{\text{INDUCE}}$ can be token sequence "`</think> The answer is \boxed`". This allows for an ongoing assessment of the model's conclusion.

**Stability-Based Stopping Criterion.** Through this criterion, the early exit mechanism stops the CoT when the model produces the same intermediate answer for $k$ consecutive steps (Fu et al.,

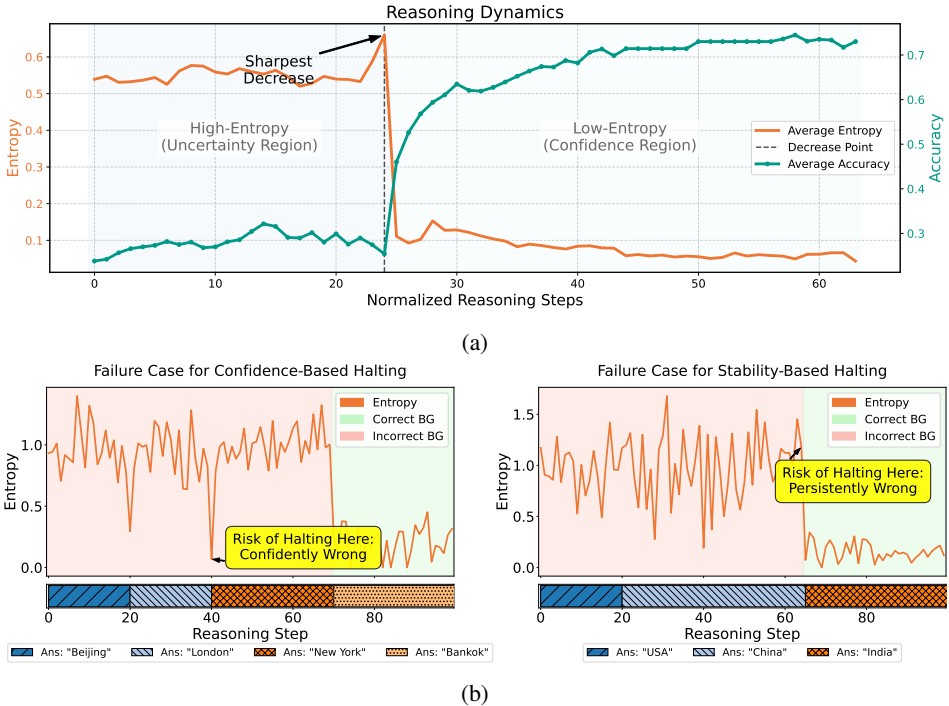

(a)

(b)

Figure 1: **Dynamics of Entropy and Accuracy across Reasoning Stages, and Illustrative Failure Modes. (a) Entropy and Accuracy Dynamics:** This plot illustrates the typical trajectory of probing entropy and corresponding accuracy over reasoning steps. It clearly delineates two distinct regions: an initial Uncertainty Region characterized by high entropy and low accuracy, followed by a Confidence Region where entropy is low and accuracy consistently stabilizes at a high level. The transition between these regions is marked by a sharp drop in entropy. **(b) Illustrative Halting Failure Modes:** This panel presents two representative cases in confidence-based halting and stability-based halting. Green (red) background means correct (wrong) answer, the bottom bar means the probed answer during reasoning steps. Case 1 shows a scenario where the model exhibits momentary low entropy spike (high confidence), leading to a confidently incorrect early exit if solely relying on a single-step confidence. Case 2 depicts a situation where the model's answer stabilizes for several steps (high entropy for the answer), yet the answer is persistently incorrect, highlighting the need for robust confidence criteria beyond mere answer stability.

2024). Formally, generation is terminated at step $i$ if:

$$\text{stop if } A_i = A_{i-1} = \cdots = A_{i-k+1}. \tag{2}$$

**Confidence-Based Stopping Criterion.** Alternatively, one could also halts the chain-of-thought as soon as the model's predictive confidence in the intermediate answer exceeds a pre-defined threshold, thereby avoiding redundant reasoning steps. To quantify the model's certainty about this intermediate answer, the most common approach is to calculate its entropy, $H(A_i, |X, T_{1:i})$. A lower entropy value signifies higher confidence. Formally, if the answer $A_i$ consists of m tokens $(a_{i,1}, \ldots, a_{i,m})$, its entropy is defined as the average token-level negative log-likelihood:

$$H(A_i|X, T_{1:i}) = -\frac{1}{m} \sum_{j=1}^{m} \sum_{v \in V} P_j(v|\cdot) \log_2 P_j(v|\cdot), \tag{3}$$

where V is the vocabulary and $P_j(v|\cdot) = P(v|X, T_{1:i}, P_{\text{INDUCE}}, a_{i,1:j-1})$ is the probability of token $v$ conditioned on all preceding text. This entropy value, $H(A_i, |X, T_{1:i})$ represents the model's uncertainty regarding its answer after $i$ steps of reasoning. (Other metrics based on normalized probabilities or geometric means also exist (Yang et al., 2025; Huang et al., 2025)).

This approach terminates generation as soon as the confidence of an intermediate probed answer $A_i$ exceeds a predefined threshold. Using entropy $H(A_i|X, T_{1:i})$ as an inverse measure of confidence,

the halting condition is:

$$\text{stop if } H(A_i|X, T_{1:i}) \leq \tau_{conf}, \tag{4}$$

where $\tau_{conf}$ is a predefined threshold. Methods like DEER (Yang et al., 2025) and the work by Huang et al. (2025) exemplify this strategy.

## 2.2 LIMITATIONS OF HEURISTIC STOPPING CRITERIA

While these heuristics represent important initial steps toward improving CoT efficiency, their reliance on localized signals—specifically, single-step confidence or answer stability—introduces fundamental limitations. These shortcomings arise because such signals often fail to reliably indicate the true completion and soundness of the underlying reasoning process. Specifically, for confidence-based stopping criterion, the core assumption is that a high-confidence score (low entropy) directly correlates with the a complete reasoning. This heuristic, however, is susceptible to "premature confidence," where a model exhibits a momentary spike in confidence for an answer that is plausible but ultimately flawed upon further deliberation as shown in the first case of Figure 1b. For stability-based stopping criterion, the underlying assumption is that stability indicates convergence to a final solution. Yet, this heuristic can be unreliable. The unreliability of this heuristic stems from its inability to consider the model's confidence. In the initial analysis phase, for example, a model might produce a sequence of same, high-entropy (uncertain) answers. A stability-based stopping criterion on its own cannot differentiate between a model converging on a well-reasoned, low-entropy answer and one that is simply stuck repeating an early, high-entropy guess as shown in the second failure case of Figure 1b.

## 3 METHOD: CONFIDENCE REGION DETECTION AND EARLY EXIT OF CoT

This paper presents the first systematic characterization of the global dynamics underlying the CoT reasoning process in LLMs. While prior work has primarily employed entropy as a local, stepwise confidence score, our work departs from this granular view. We introduce a global analysis that captures the full temporal progression of entropy, enabling us to move beyond transient fluctuations and toward a principled understanding of how LLMs converge on a solution. This holistic perspective is the key to delineating distinct phases of reasoning and identifying the precise points of commitment to a final answer.

Table 1: Statistical Analysis of Halting Signals in High and Low-Entropy Regions across 100 Samples.

| Reasoning Region | Low-Entropy Points | | | Stable Answer Sequences | | |
|---|---|---|---|---|---|---|
| | Total | Correct | Accuracy | Total | Correct | Accuracy |
| High-Entropy Region | 33 | 3 | 9.0% | 65 | 17 | 26.2% |
| Low-Entropy Region | 72 | 43 | 59.7% | 65 | 48 | 73.8% |

### 3.1 CONFIDENCE REGION OF CoT

To quantitatively track the LLM's reasoning state, we monitor the dynamics of its predictive entropy throughout the generation process. First, we sample 100 instances from the Bespoke-Stratos-17k dataset (Labs, 2025), a heterogeneous collection spanning domains such as coding, mathematics, science, and logic puzzles, to ensure the diversity of our analysis. An initial observation across these samples reveal a consistent pattern in their entropy evolution: an initial phase of high entropy, followed by a sharp decrease, and finally settling into a low-entropy plateau (see Figure 7 in Appendix for individual examples). To visualize this universal phenomenon in aggregate, we present the average entropy dynamics for Qwen3-4B-Thinking-2507 in Figure 1a. To enable a meaningful comparison across samples of varying lengths, we normalize the temporal axis of each reasoning process. We identify the point of sharpest entropy decrease as a critical point for each sample. Then, using linear interpolation, we align all samples at this decrease point and normalize the lengths of the high-entropy (pre-transition) and low-entropy (post-transition) phases, respectively.

As illustrated in Figure 1a, this aggregation clearly reveals a distinct two-region structure within the CoT process, connected by a sharp drop in entropy. We analyze the characteristics of high- and low-entropy regions by evaluating the accuracy of prior early-exit methods in each phase, with a statistical summary provided in Table 1. Our key findings are as follows:

- Probing answers in the high-entropy region are unreliable. This is visually corroborated by Figure 1a, which demonstrates that the overall probing accuracy remains consistently low throughout this phase. Furthermore, our analysis of low entropy points and stable answer sequences in this region (Table 1) reveals that confidence-based halting achieves an accuracy of only 9.0%, while stability-based halting is correct merely 26.2% of the time.

- Probing answers in the low-entropy region are highly reliable. In stark contrast, once the process transitions into the low-entropy region where Figure 1a clearly shows the accuracy plateauing at a high level. Furthermore, our analysis of low entropy points and stable answer sequences in this region (Table 1) reveals that the accuracy of confidence-based halting jumps to 59.7%, and the accuracy of stability-based halting reaches 73.8% (Table 1).

- The low-entropy region is often excessively long. Although the correct answer is typically found early in this phase, Figure 1a shows that the model continues to generate a substantial number of tokens in this confident state. This post-solution verbosity is the primary source of redundancy that our method aims to eliminate.

Based on these observations, we posit that the high-entropy phase corresponds to the model's **Uncertainty Region**, whereas the low-entropy phase represents its **Confidence Region**. The confluence of reliable answers in the Confidence Region, unreliable answers in the Uncertainty Region, and the inherent verbosity of the Confidence Region leads to our central conclusion: *a superior strategy is to perform an early exit precisely when the model has transitioned into the Confidence Region, while explicitly avoiding premature exits from the Uncertainty Region.*

## 3.2 CORE: CONFIDENCE REGION EXIT

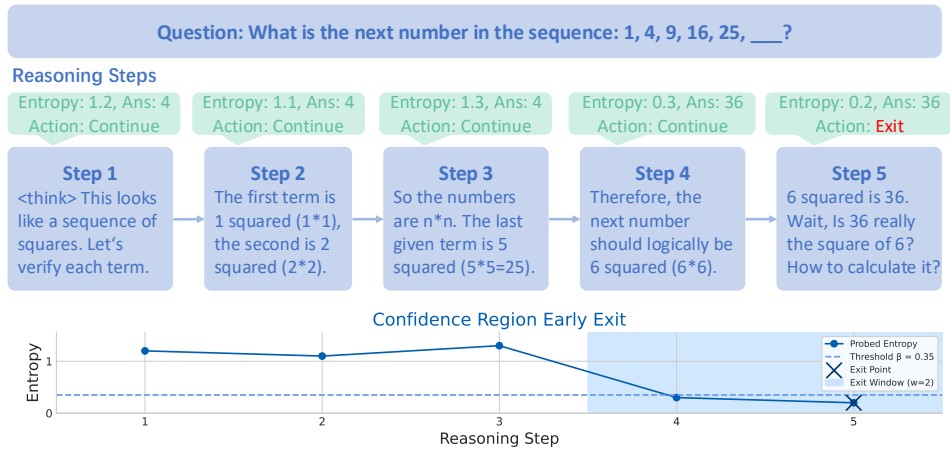

Figure 2: This figure illustrates the step-by-step operation of the CORE algorithm using a sample reasoning trace. The top panel shows the model's generated answer, probed entropy, and the CORE algorithm's decision ("Continue" or "Exit") at each step. The bottom panel visualizes the corresponding entropy curve over reasoning steps, along with the predefined confidence threshold ($\beta = 0.35$) and window size ($w = 2$). The algorithm effectively identifies and triggers an early exit only when the entropy consistently falls below the threshold within the specified window, thereby avoiding premature exits based on spurious confidence dips.

While it is straightforward to distinguish between the uncertainty and confidence regions based on sharply decread point when the entire Chain of Thought (CoT) is known, this is not feasible during inference as the global entropy landscape is unavailable. Consequently, identifying the transition

into the confidence region using relative entropy is challenging. To address this, we propose a method that utilizes a threshold and a sliding window to determine whether the model has entered the confidence region, triggering an early exit if it has.

At each generation step $i$, the algorithm first determines the current reasoning region, $S_i$. This is achieved by monitoring the probed entropy signal, $H(A|X, T_{1:i})$, which reflects the model's certainty about the final answer. Based on a predefined threshold $\beta$, the stage is identified as:

$$S_i = \begin{cases} \text{Confidence Region} & \text{if } \frac{\sum_{\max(1, i-w+1)}^{i} H(A|X, T_{1:j})}{\max(i, w)} \leq \beta, \\ \text{Uncertainty Region} & \text{otherwise,} \end{cases} \quad (5)$$

Once the chain-of-thought enters the Confidence Region, generation is immediately halted and the current trial answer is emitted as the final prediction.

## 4 EXPERIMENTS

### 4.1 EXPERIMENTAL SETUP

**Setup** Our experiments utilize three representative open-source LLMs of varying sizes: DeepSeek-R1-Distill-Qwen-7B, Qwen3-4B-Thinking-2507, and Qwen3-14B. We evaluate these models on four challenging reasoning benchmarks: AIME24, AIME25, and GPQA-Diamond (Rein et al., 2024). For all models, we adhere to their standard prompting strategies, with full details provided in Figure 6 of the appendix. To ensure robust and stable performance estimates, we vary the number of evaluation runs based on the dataset size. For the large-scale GPQA-Diamond benchmark, we report results averaged over 4 random runs. For the smaller competition math datasets (AIME24 and AIME25), we perform 16 random runs per dataset to mitigate the effects of sampling variance.

**Hyperparameter Settings** For each model, we tune hyperparameters on the Bespoke-Stratos-17k dataset. These optimized settings were then applied directly to the test datasets for final evaluation. Detailed hyperparameter configurations are provided in the Appendix C.1.

**Baselines** We compare our method with two representative early exit method: DEER (Yang et al., 2025), Dynasor (Fu et al., 2024).

Table 2: Main results on aggressive early exit.

| Methods | AIME25 | | AIME24 | | GPQA | | Average | |
|---|---|---|---|---|---|---|---|---|
| | Acc ↑ | Tokens ↓ | Acc ↑ | Tokens ↓ | Acc ↑ | Tokens ↓ | Acc ↑ | Tokens ↓ |
| *DeepSeek-R1-Distill-Qwen-7B* | | | | | | | | |
| Vanilla | 41.04 | 14556 | 55.63 | 13313 | 38.38 | 8637 | 45.02 | 12169 |
| DEER | 35.21 | 10934 | 45.2 | 9858 | 30.05 | 6104 | 36.82 | 8965 |
| Dynasor | 32.22 | 10863 | 49.33 | 9761 | 17.4 | 6239 | 32.98 | 8954 |
| Ours | **36.46** | 10965 | **51.25** | 9540 | **42.80** | 6243 | **43.50** | 8916 |
| *Qwen3-4B-Thinking-2507* | | | | | | | | |
| Vanilla | 81.04 | 22613 | 77.29 | 19178 | 64.65 | 9442 | 74.32 | 17078 |
| DEER | 64.32 | 16925 | 66.97 | 14348 | 61.5 | 6600 | 64.26 | 12624 |
| Dynasor | **65.02** | 16930 | 66.32 | 14623 | **62.38** | 6463 | 64.57 | 12672 |
| Ours | 63.75 | 16953 | **68.33** | 14554 | 62.34 | 6421 | **64.81** | 12643 |
| *Qwen3-14B* | | | | | | | | |
| Vanilla | 71.67 | 16727 | 81.46 | 13872 | 67.93 | 5988 | 73.69 | 12196 |
| DEER | 55.0 | 11043 | 62.0 | 9003 | **63.64** | 3854 | 60.21 | 7967 |
| Dynasor | 54.25 | 11231 | 62.12 | 9032 | 59.09 | 3926 | 58.49 | 8063 |
| Ours | **60.63** | 10958 | **65.21** | 8980 | **63.64** | 4192 | **63.16** | 8043 |

Table 3: Main results on conservative early exit.

| Methods | AIME25 | | AIME24 | | GPQA | | Average | |
|---|---|---|---|---|---|---|---|---|
| | Acc ↑ | Tokens ↓ | Acc ↑ | Tokens ↓ | Acc ↑ | Tokens ↓ | Acc ↑ | Tokens ↓ |
| *DeepSeek-R1-Distill-Qwen-7B* | | | | | | | | |
| Vanilla | 41.04 | 14556 | 55.63 | 13313 | 38.38 | 8637 | 45.02 | 12169 |
| DEER | 40.0 | 13508 | 50.62 | 11184 | 35.73 | 7227 | 42.12 | 10639 |
| Dynasor | 37.78 | 13544 | 52.67 | 11215 | 19.32 | 7342 | 36.59 | 10700 |
| Ours | **40.83** | 13634 | **55.0** | 11806 | **40.53** | 7160 | **45.45** | 10867 |
| *Qwen3-4B-Thinking-2507* | | | | | | | | |
| Vanilla | 81.04 | 22613 | 77.29 | 19178 | 64.65 | 9442 | 74.32 | 17078 |
| DEER | 74.17 | 20384 | 72.67 | 17272 | 62.84 | 8450 | 69.89 | 15369 |
| Dynasor | 73.33 | 20362 | 73.0 | 17125 | **64.65** | 8367 | 70.33 | 15285 |
| Ours | **76.25** | 20164 | **74.38** | 17270 | 64.27 | 8492 | **71.63** | 15309 |
| *Qwen3-14B* | | | | | | | | |
| Vanilla | 71.67 | 16727 | 81.46 | 13872 | 67.93 | 5988 | 73.69 | 12196 |
| DEER | 68.5 | 14424 | 70.42 | 11454 | 61.87 | 5368 | 66.93 | 10415 |
| Dynasor | 68.67 | 14305 | 71.56 | 11866 | 61.74 | 5317 | 67.32 | 10496 |
| Ours | **68.75** | 14351 | **76.04** | 11358 | **65.91** | 5123 | **70.23** | 10277 |

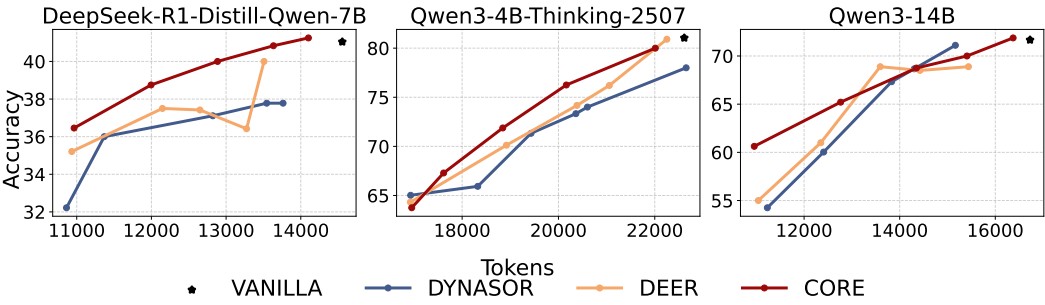

Figure 3: Pareto-frontier on AIME25 dataset.

## 4.2 BETTER EFFICACY-EFFICIENCY TRADE-OFF

Given the trade-off between accuracy and generated tokens, comparing Pareto-optimal frontiers is an effective evaluation method. To generate points along these frontiers, we vary the number of tokens by adjusting key hyperparameters for each method: the confidence threshold for DEER, the stable answer threshold for Dynasor, and the window size for CORE. For computational efficiency, we focus our Pareto frontier analysis on the most challenging dataset, AIME25. On other datasets, we evaluate model performance at two distinct operational points: an aggressive early exit setting targeting a 25% token reduction, and a conservative early exit setting with a 10% token reduction. The results are presented in Tables 2 and 3.

In aggressive early exit scenarios (Table 2), CORE proves highly efficient compared to previous methods. For instance, on average with the DeepSeek-R1-Distill-Qwen-7B model, it achieves 43.50% accuracy using only 8916 tokens, outperforming both DEER (36.82%) and Dynasor (32.98%). When more computational budget is allowed (Table 3), CORE not only delivers the highest accuracy but also reduces token usage by approximately 10% compared to the Vanilla baseline for comparable performance. This unique adaptability in managing the accuracy-cost trade-off holds consistently across all tested models, proving CORE is a robust and effective inference optimization framework. As illustrated in Figure 3, CORE clearly achieves a superior Pareto frontier compared to the other methods.

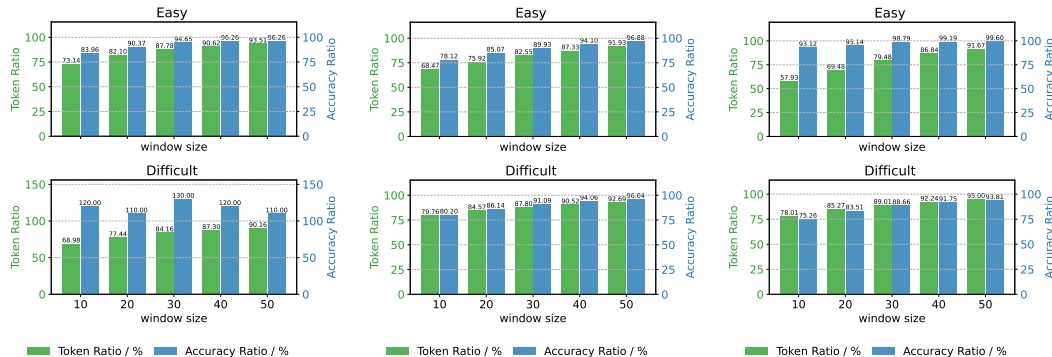

(a) Performance on DeepSeek-R1-Distill-Qwen-7B.

(b) Performance on Qwen3-4B-Thinking-2507.

(c) Performance on Qwen3-14B Model.

Figure 4: Performance of CORE on the AIME25 dataset across different problem difficulties and model sizes. The charts show CORE's token and accuracy ratios relative to a vanilla baseline. For easy problems, CORE demonstrates token reduction with a much smaller sacrifice in accuracy across all models. For difficult problems, the method's effectiveness varies by model architecture: on DeepSeek-R1-Distill-Qwen-7B model, it simultaneously delivers dramatic token savings while significantly boosting accuracy. In contrast, on the more powerful Qwen3 models, the token savings are accompanied by a roughly proportional decrease in accuracy.

## 4.3 ANALYSIS ON DIFFERENT DIFFICULTY LEVELS

To investigate our method's performance, we analyze its accuracy and token consumption across problems categorized as 'easy' or 'difficult' based on the model's vanilla inference accuracy on AIME25 dataset. We then evaluate CORE's performance ratios against the vanilla baseline as a function of increasing window size in Figure 4. For easy problems, CORE demonstrates token reduction with a much smaller sacrifice in accuracy across all models. This efficiency gain is notably prominent in Qwen3-14B, demonstrating CORE's effectiveness in easy problems. On difficult problems, CORE's impact varies by model: for the distilled DeepSeek-R1-Distill-Qwen-7B, it dramatically saves tokens while concurrently achieving a significant accuracy boost, effectively pruning error accumulation from misguided multi-step explorations. Conversely, for the more powerful Qwen3 models, token savings are generally met with a proportional accuracy decrease, suggesting their vanilla reasoning on difficult problems is already more optimized and contains less redundancy.

## 4.4 COST ANALYSIS

A potential drawback of probing is the introduction of additional inference latency, which might partially offset the efficiency gains achieved by shortening CoT sequences. To precisely quantify this trade-off, we conduct an empirical analysis on the Qwen3-4B-Thinking-2507 model using the AIME25 dataset. We compare the total inference time for scenarios employing our conservative and aggressive early exit strategies against the vanilla CoT baseline. As depicted in Figure 5, our analysis reveals that despite the inherent overhead associated with probing, the additional inference latency introduced by CORE is remarkably minimal. This strongly suggests that the substantial efficiency gains derived from reduced token

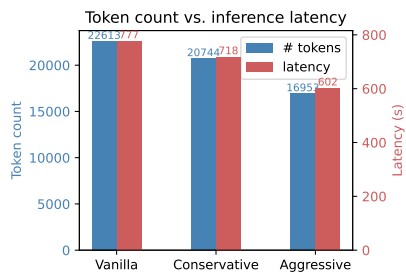

Figure 5: Cost analysis.

usage are largely preserved, making our method a net positive in terms of overall runtime.

## 5 RELATED WORK

Improving the efficiency of Chain-of-Thought (CoT) reasoning is a critical area of research. Previous work can be categorized into training-based and training-free approaches. Our work builds upon the latter. In this section, we provide a brief review of previous methods.

### 5.1 TRAINING-BASED METHODS

Training-based methods seek to enhance efficiency by fine-tuning a model's parameters to produce more concise reasoning. These approaches generally fall into three main categories. One major line of work focuses on representation, compressing reasoning steps into a continuous latent space rather than explicit text (Continuous CoT) (Hao et al., 2024; Zhang et al., 2025b; Shen et al., 2025). Another prominent strategy involves Supervised Fine-Tuning (SFT), where models are trained on datasets of curated, concise CoTs, often distilled from longer, more verbose examples (Yu et al., 2025; 2024; Kang et al., 2025). A third category utilizes Reinforcement Learning (RL) to explicitly incentivize brevity, typically by incorporating length-based penalties into the reward function or using preference optimization to favor shorter, correct answers (Tu et al., 2025; Li et al., 2025; Dai et al., 2025; Xu et al., 2025b; Yu et al., 2025; Hou et al., 2025; Zhang et al., 2025a; Liu et al., 2025; Arora & Zanette, 2025).

### 5.2 TRAINING-FREE METHODS

Training-free methods offer a lightweight, model-agnostic alternative for dynamic CoT termination by employing various heuristics. Some methods are confidence-based, halting generation when a single answer's confidence is high (Yang et al., 2025). Others are stability-based, terminating when an intermediate answer is repeated consecutively (Fu et al., 2024). A third approach focuses on exploration suppression; rather than directly halting, it discourages further deliberation by reducing the probability of "exploration" tokens when the model is already confident (Huang et al., 2025). Finally, another category of methods uses prompt engineering to elicit more succinct answers from the model (Xu et al., 2025a). We consider such approaches to be model-specific and thus do not explore them further in this work. However, we argue that the aforementioned halting-based approaches predominantly rely on heuristic criteria and lack a principled understanding of the CoT process.

## 6 CONCLUSIONS

In this paper, we introduce a novel, entropy-driven early exit framework designed to enhance the efficiency of Chain-of-Thought (CoT) reasoning in Large Language Models. Our foundational insight stems from pioneering an empirical investigation into the dynamic evolution of entropy during the CoT process. We conclusively demonstrate a consistent pattern: CoT generation transitions from an initial high-entropy uncertainty region, characterized by exploration, to a stable, low-entropy confidence region, signaling the convergence of reasoning. Leveraging this robust correlation between the entry into the low-entropy confidence region and the completion of a sound reasoning process, we propose a lightweight, training-free halting criterion. This principled approach allows models to terminate generation precisely when reasoning is concluded, effectively avoiding both premature exits that compromise accuracy and redundant verbosity that inflates computational costs. Our comprehensive experiments across diverse LLMs (DeepSeek-R1-Distill-Qwen-7B, Qwen3-4B-Thinking-2507, and Qwen3-14B) and challenging benchmarks (AIME24, AIME25, GPQA) affirm the efficacy of our method. We demonstrate that our approach achieves a superior Pareto-optimal trade-off between computational cost and accuracy when compared to existing training-free early exit methods. This work not only offers a novel, information-theoretic perspective for understanding the internal dynamics of the CoT process but also provides a more principled and robust method for efficient reasoning. We believe these insights will pave the way for developing more economically viable and performant LLMs, fostering further research into the inherent predictability and controllability of complex reasoning patterns.

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

## A  THE USE OF LLMS

In the preparation of this manuscript, LLMs were utilized as a writing assistant to enhance clarity, refine phrasing, and improve overall readability. Specifically, LLM tools were employed for:

- Grammar and Style Refinement: Identifying and correcting grammatical errors, improving sentence structure, and suggesting more concise or academic phrasing.
- Vocabulary Enhancement: Proposing alternative word choices to avoid repetition and enrich the lexical diversity of the text.
- Conciseness: Restructuring sentences or paragraphs to ensure that arguments are presented in a clear and succinct manner.

It is important to note that LLMs were used solely for **editorial support and linguistic enhancement**. All research ideas, experimental design, data analysis, interpretation of results, and the core scientific content presented in this paper are the original work of the authors. The LLM's role was strictly limited to improving the articulation of these original contributions, not to generate any scientific content or insights. The authors have meticulously reviewed and approved all LLM-assisted revisions to ensure accuracy and alignment with their intended meaning.

## B  ETHICS STATEMENT

This research aims to enhance the computational efficiency and understanding of LLM reasoning. All experiments utilized publicly available, non-sensitive benchmark datasets (AIME24, AIME25, GPQA) and pre-trained models within their respective licenses.

We acknowledge the potential for dual-use concerns, as increased efficiency could inadvertently lower the barrier for malicious LLM applications, such as large-scale misinformation generation.

However, the primary impact of our work is to significantly reduce computational overhead, thereby promoting more environmentally sustainable AI and democratizing access to advanced LLM reasoning for researchers with limited resources. We are committed to responsible AI development and believe our contributions offer clear positive value to the community.

## C  REPRODUCIBILITY STATEMENT

To facilitate the reproducibility of our findings, we provide a detailed account of our methodology, implementation, and experimental setup. The core logic and formulation of our proposed method, CORE, are described in Section 3. For implementation-specific details, the exact prompts used for different dataset is in Figure 6 in the Appendix. A comprehensive list of all hyperparameters, including the window size ($w$) and confidence threshold ($\beta$) used for each model, is provided in Appendix C.1. All datasets used in our evaluation (AIME25, AIME24, and GPQA) are publicly available. To ensure the stability and robustness of our findings, all experiments were repeated multiple times, and the averaged results are reported. Our source code will be made publicly available upon acceptance of this manuscript.

### C.1  HYPER-PARAMETER SETTINGS

Our approach introduces only two fine-tuned hyper-parameters: the entropy threshold $\beta$ and the sliding-window size $w$. We tune $\beta$ directly on the Bespoke-Stratos-17k training split and show the final decision in Table 4. For $w$, we grid-search the values [10,20,30,40,50] and select the two settings whose CoT early-exit rates fall closest to 75% and 85%; these two checkpoints are reported as our main experimental results. Then we plot the pareto frontier for AIME25 using all the five points.

## D  EXPERIMENTS

### D.1  EMPIRICAL VISUALIZATION OF CoT

Table 4: Hyper-parameter settings.

| Model | $\beta$ |
|---|---|
| DeepSeek-R1-Distill-Qwen-7B | 0.2 |
| Qwen3-4B-Thinking-2507 | 0.05 |
| Qwen3-14B | 0.02 |

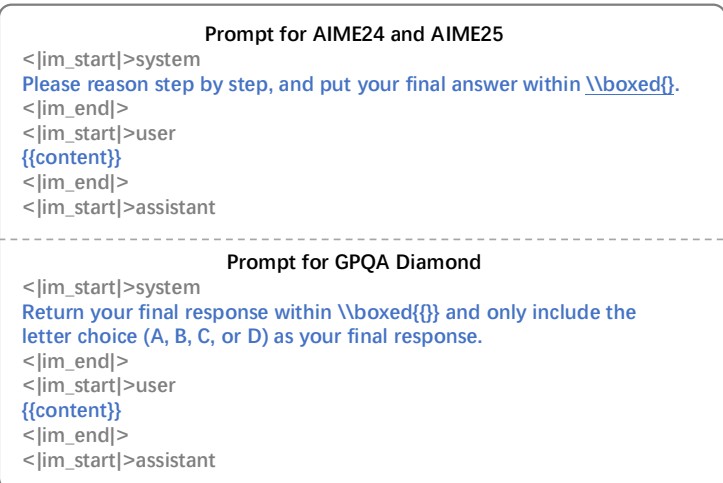

**Prompt for AIME24 and AIME25**
<|im_start|>system
Please reason step by step, and put your final answer within \\boxed{}.
<|im_end|>
<|im_start|>user
{{content}}
<|im_end|>
<|im_start|>assistant

- - - - - - - - - - - - - - - - - - - - - - - - - - - - - - - - - - - -

**Prompt for GPQA Diamond**
<|im_start|>system
Return your final response within \\boxed{{}} and only include the letter choice (A, B, C, or D) as your final response.
<|im_end|>
<|im_start|>user
{{content}}
<|im_end|>
<|im_start|>assistant

Figure 6: Prompt for different dataset.

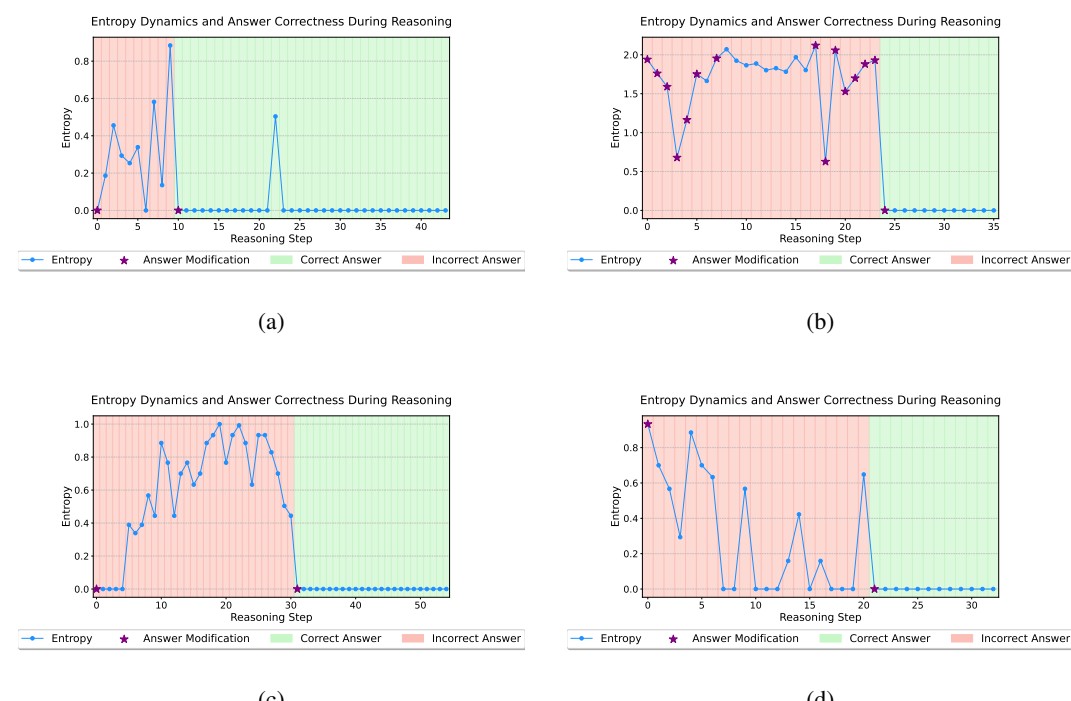

(a)

(b)

(c)

(d)

Figure 7: An illustration of reasoning dynamics across four different scenarios. In each subplot, the **blue line** tracks the model's entropy (y-axis) at each reasoning step (x-axis). **Background shading** indicates the correctness of the intermediate answer when compared to the ground truth (green for correct, red for incorrect). **Purple stars** mark the exact steps where the model modifies its answer. The subplots show: (a) (b) (c) shows that stopping at high confidence (low entropy) may lead to uncomplete answer . (c) and (d) also shows that premature convergence on an incorrect answer.