# OpenReview forum: "Probing Confidence Regions for Early Exits in Chain-of-Thought"
_ICLR.cc/2026/Conference — Submitted to ICLR 2026_

### Official Review · Reviewer_EDuu · 2025-10-31

**Soundness:** 2
**Presentation:** 3
**Contribution:** 2
**Rating:** 2
**Confidence:** 4

**Summary:**

This paper proposes CORE, a training-free early-exit framework for CoT reasoning in large language models. The key idea is to monitor the entropy dynamics of intermediate reasoning steps, observing that CoT generation typically transitions from a high-entropy uncertainty region to a low-entropy confidence region. CORE detects this transition using a sliding-window average of entropy and halts generation once the model stably enters the low-entropy region, aiming to reduce redundant reasoning without hurting accuracy.

**Strengths:**

1. Clear empirical observation of entropy dynamics. The paper provides a clean and consistent empirical finding that CoT reasoning trajectories often exhibit a transition from high-entropy exploration to low-entropy convergence, which is intuitively interpretable and empirically supported by multiple examples.

2. Simple yet effective method. The proposed CORE mechanism is lightweight, training-free, and easy to implement across different LLMs. It offers a practical solution for improving reasoning efficiency without requiring additional model fine-tuning.

**Weaknesses:**

1. Heuristic nature of the method. Despite the appealing intuition, the proposed CORE remains largely a heuristic extension of existing confidence-based early-exit methods. The confidence region is defined purely through an empirical entropy threshold and sliding window, without theoretical grounding or formal justification that low entropy reliably corresponds to reasoning completeness.

2. The entropy threshold (β) and window size (w) are tuned empirically on a specific dataset (Bespoke-Stratos-17k) and directly applied to test tasks. This raises concerns about generalization and stability across domains or reasoning styles.

3. The main empirical observation that CoT entropy decreases as reasoning progresses is not entirely new. Similar dynamics have been reported in recent works [1,2]. The proposed framework essentially repackages this known pattern rather than providing new methodological or conceptual insights. The proposed confidence region essentially monitors whether the token-level entropy remains below a threshold β for w consecutive steps. In other words, it merely adds a smoothing window on top of standard confidence-based early-exit heuristics

4. The baselines (DEER, Dynasor) are relatively early heuristic methods. The paper does not compare against recent adaptive or RL-based approaches such as ThinkPrune[1], SelfBudgeter[2], which limits the strength of the empirical claims.



[1] Hou, B., Zhang, Y., Ji, J., Liu, Y., Qian, K., Andreas, J., & Chang, S. (2025). Thinkprune: Pruning long chain-of-thought of llms via reinforcement learning. arXiv preprint arXiv:2504.01296.

[2]Li, Z., Dong, Q., Ma, J., Zhang, D., Jia, K., & Sui, Z. (2025). Selfbudgeter: Adaptive token allocation for efficient llm reasoning. arXiv preprint arXiv:2505.11274.

**Questions:**

See weaknesses

---

> ### Author Response · Authors · 2025-11-21
> **Response to Reviewer EDuu**
>
> We thank you for your time and feedback. We appreciate that the reviewer found our empirical observation "clear and consistent" and our method "simple yet effective" and "practical."
> We address your concerns as follows.
>
> **Q1. Regarding the Heuristic Nature**
>
> A1: We acknowledge that CORE is primarily empirically grounded. Our key contribution is the discovery of the "Uncertainty Region $\to$ Confidence Region" dynamic. Based on this, we propose that reasoning completion is a **stable state** rather than a point, and CORE (using window $w$) is designed to capture this stability. While theoretical analysis of LLM internal states remains an open challenge due to their black-box nature, our extensive experiments across diverse mainstream models confirm that this two-phase pattern is a robust and universal phenomenon, providing strong empirical justification for our approach. Theoretical grounding is an exciting direction for future work.
>
> **Q2. Regarding the Hyperparameter generalization and stability**
>
> A2: We calibrated $\beta$ and $w$ using only **100 random samples** from a single validation set (Bespoke-Stratos-17k) and applied these **identical** hyperparameters across all our diverse benchmarks (Math, QA). The fact that these settings transfer robustly to completely different tasks and datasets demonstrates that CORE captures a general property of the reasoning process, rather than overfitting to a specific domain.
>
> **Q3. Similar entropy dynamics have been reported in recent works [1,2]. The proposed framework is limited in methodological or conceptual insights.**
>
> A3: First, regarding the novelty of the entropy dynamics: The cited works [1,2] focus on RL-based pruning and token allocation, but do not investigate the specific entropy dynamics of the reasoning process. To the best of our knowledge, we are the first to systematically characterize the global "Uncertainty $\to$ Confidence" transition.
>
> Second, regarding the algorithmic contribution: We understand why the simplicity of using a sliding window might be perceived as incremental. However, we respectfully argue that the main contribution is conceptual.
> - **Conceptual Leap:** The key failure mode of prior work (e.g., DEER, which is effectively CORE with $w=1$) is that it is susceptible to premature confidence (as shown in Figure 1a). This is because it incorrectly assumes reasoning completion is an instantaneous point.
> - **Our Contribution:** Our core scientific discovery is that reasoning completion is not a point, but a **stable state** (the "Confidence Region"). The sliding window ($w>1$) is not just "signal smoothing"; it is the algorithmic operationalization of this new concept of "stability." This conceptual leap—from "point" to "stable state"—is what allows our method to solve the premature exit problem that simpler methods cannot.
>
> **Q4. Comparison with RL-based methods**
>
> A4: CORE is a training‑free, plug‑and‑play early‑exit mechanism. The cited works [1,2] are RL‑based methods that require substantial training and computational resources. As noted in lines 14–15 and 48–58 of the paper, our approach is fundamentally different: we aim to provide a lightweight, inference-time solution that does not require modifying the model or expensive training. Therefore, they are not directly comparable baselines for our specific problem setting.
>
> [1] Hou, B., Zhang, Y., Ji, J., Liu, Y., Qian, K., Andreas, J., & Chang, S. (2025). Thinkprune: Pruning long chain-of-thought of llms via reinforcement learning. arXiv preprint arXiv:2504.01296.
>
> [2]Li, Z., Dong, Q., Ma, J., Zhang, D., Jia, K., & Sui, Z. (2025). Selfbudgeter: Adaptive token allocation for efficient llm reasoning. arXiv preprint arXiv:2505.11274.

---

### Official Review · Reviewer_7NGG · 2025-10-31

**Soundness:** 3
**Presentation:** 3
**Contribution:** 2
**Rating:** 4
**Confidence:** 4

**Summary:**

This paper introduces CORE (COnfidence Region Exit), a training-free method to improve the efficiency of Chain-of-Thought (CoT) reasoning in large language models. The authors observe that a model’s reasoning entropy follows a consistent two-phase pattern: an initial high-entropy “uncertainty region” where answers fluctuate and are unreliable, followed by a low-entropy “confidence region” where reasoning stabilizes and accuracy plateaus. Building on this finding, CORE detects entry into the confidence region using a sliding-window entropy threshold and halts generation once stability persists, thus avoiding both premature exits and redundant post-solution verbosity. The experiments show that CORE has better effectiveness and efficiency trade-off compared with the baselines.

**Strengths:**

1. Present a clear empirical observation of reasoning dynamics, especially the two-phase entropy pattern, quite interesting.

2. The method itself is simple and training-free.

3. Experiments on diverse tasks and models shows the generality of the method.

4. Easy to follow.

**Weaknesses:**

1. It seems the proposed method relies on a hidden assumption: the model is calibrated well. If the model is poorly calibrated, the model can be easily trapped in a low-entropy region but with wrong answers. This paper does not have sufficient discussion on this.

2. This work seems to only discuss the short CoT cases, i.e., only one reasoning chain is generated. Reasoning models like Deepseek-R1 or other models that involve test-time scaling, can generate multiple CoT chains separated by tokens like "wait". In this case, there may exist a transition from the low-confidence region into the high-entropy region back and forth. In this case, how to decide when to exit using the confidence region still needs consideration.

3. While the proposed method can improve the effectiveness, it is a simple extension of identifying the low entropy point into identifying a low entropy period. The proposed method seems to be incremental. Besides, the thresholding method (for both thresholds and window size) lacks sufficient discussion. The current hyperparameters are determined on an auxiliary dataset, and perhaps a better or more principal tuning method could be discussed.

**Questions:**

See weakness.

---

> ### Author Response · Authors · 2025-11-21
> **Response to Reviewer 7NGG**
>
> Thank you for your feedback and for finding our work "interesting," "simple," "general," and "easy to follow." We appreciate the 3/3 "good" ratings for Soundness and Presentation. We will address the weaknesses in turn.
>
> **Q1: Regarding the "Hidden Assumption" of Model Calibration.**
>
> A1: We focus on mainstream, well-calibrated models. To address the concern about the model being "trapped" in a low-entropy region with a wrong answer ("confident failure"), we compared CORE with the vanilla baseline.
>
> | Method | Confident Failures (accuracy / # tokens) |
> | :--- | :--- |
> | Vanilla | 0 / 20,717 |
> | **CORE** | **0 / 20,056** |
>
> **Interpretation:** In cases of "confident failure" (where the model is certain but wrong), vanilla decoding continues to generate a long chain only to output the same incorrect answer. CORE detects this high confidence and exits early. Crucially, CORE **does not alter the final outcome** (it remains incorrect, just as in vanilla), but it reduces the computational cost. Thus, even for poorly calibrated predictions, CORE improves efficiency without degrading accuracy relative to the baseline.
>
> **Q2: Applicability to long CoT and test-time scaling models (e.g., DeepSeek-R1).**
>
> A2: We clarify that our experiments **do** cover such scenarios. We utilized the **DeepSeek-R1-Distill-Qwen-7B** model (table 2 and table 3). This model frequently generates long CoT responses interspersed with "wait" tokens. For instance, on the AIME25 dataset, it produces an average of 10,965 tokens per solution, containing 63 "wait" tokens—confirming we are testing genuine long-horizon reasoning with multiple chain separations. Despite these complex reasoning patterns, our empirical results (Figure 1) consistently show the same global dynamic: an initial high-entropy uncertainty region followed by a stable low-entropy confidence region.
>
> **Q3: The method seems incremental.**
>
> A3: Our primary contribution is the discovery of the **"Uncertainty Region $\to$ Confidence Region"** dynamic. CORE is the algorithmic realization of this insight: we redefine reasoning completion as a **stable state** rather than a momentary point. The sliding window ($w$) is not merely a smoothing technique but a necessary proxy for detecting this stability, solving the "premature exit" failure mode of prior point-based methods.

---

### Official Review · Reviewer_YArb · 2025-10-31

**Soundness:** 2
**Presentation:** 3
**Contribution:** 2
**Rating:** 4
**Confidence:** 4

**Summary:**

This paper addresses the inefficiency of Chain-of-Thought (CoT) reasoning in LLMs. The authors identify a core dynamic: CoT processes consistently transition from a high-entropy "Uncertainty Region" to a stable, low-entropy "Confidence Region," with the latter reliably signaling reasoning completion. Based on this insight, they propose CORE (Confidence Region Exits), a simple, training-free early exit method that uses a sliding window to detect this transition and terminate generation. Extensive experiments show that CORE achieves a superior accuracy-efficiency trade-off compared to existing early exit strategies.

**Strengths:**

1. The proposed CORE method is elegant, training-free, and model-agnostic.
2. The claims are backed by rigorous and comprehensive experiments across multiple models and challenging benchmarks.

**Weaknesses:**

W1: Insufficient Analysis of Hyperparameter Sensitivity: The performance of CORE hinges on two key hyperparameters: the entropy threshold β and the window size w. The paper lacks a thorough discussion and sensitivity analysis of their selection, which leaves open questions about the method's robustness and ease of use in broader scenarios.
W2: Limited Scope of Task Domains in Evaluation: The experiments are exclusively focused on analytical reasoning tasks (math, QA). The core assumption that CoT reasoning converges to a stable low-entropy state may not hold for more open-ended, generative tasks, narrowing the demonstrated applicability of the CORE method.

W3: Under-discussed Overhead of the Probing Mechanism: The paper does not sufficiently quantify the computational overhead introduced by the "probing" step itself. A more detailed analysis of this trade-off is needed, especially for simpler problems with short reasoning chains.

W4: Failure to Distinguish Between Different Failure Modes: The paper does not analyze how CORE behaves under different types of reasoning failures, such as "confident failures" (converging to a low-entropy wrong answer) versus "lost failures" (never reaching a confidence region). This analysis is crucial for a complete understanding of the method's limitations and risk profile.

**Questions:**

Q1 (Regarding W1): Could you please provide a sensitivity analysis or ablation study for the hyperparameters β and w to help understand the method's robustness?

Q2 (Regarding W2): To what extent do you believe the core assumption of a two-phase entropy dynamic generalizes to more creative or generative CoT tasks?

Q3 (Regarding W3): Have you considered more cost-effective probing strategies, such as adaptive or sparse probing, to minimize the latency overhead?

Q4 (Regarding W4): Could you provide a breakdown of the failure cases in your experiments (e.g., "confident failures" vs. "lost failures")? This would offer a clearer picture of the method's failure modes.

---

> ### Author Response · Authors · 2025-11-21
> **Response to reviewer YArb**
>
> Thank you for your feedback. We are very pleased that you found our CORE method to be "elegant, training-free, and model-agnostic" and our experiments to be "rigorous and comprehensive."
> Here is our response to the weaknesses and questions.
>
> **Q1: Hyperparameter sensitivity analysis on $\beta$ and $w$**
>
> A1: The table below reports our method's performance on Deepseek‑R1‑Distill‑Qwen‑7B for $\beta=0.01,\,0.02,$ and $0.05$. The results indicate that the model's performance is robust to changes in $\beta$.
>
> | $\beta$ | 0.01 | 0.02 | 0.05 | 0.1 |
> |----|----:|----:|----:|----:|
> | Accuracy | 40.04 | 40.42 | 40.42 | 39.38 |
> | # tokens | 13,534 | 13,444 | 13,061 | 12,603 |
>
> For the window size $w$, we provide a detailed analysis in Figure 3 and Figure 4.
>
> **Q2: Generalizability to Generative Tasks**
>
> A2: Our method is primarily developed and evaluated on convergent reasoning benchmarks (e.g., mathematics and QA), which are the standard datasets used in this line of work. Extending CORE to open-ended generative tasks is an important and interesting direction. One practical approach would be to apply lightweight probes to a compact representation or summary of the model’s generative state (for example, probe intermediate summaries or task-specific control tokens) instead of probing the entire generated stream. We regard this extension as promising but orthogonal to the current submission, and therefore leave it for future work.
>
> **Q3a (Weakness 3): More analysis on the overhead of 'probing' and short reasoning chains**
>
> A3a: Our probing runs every 128 tokens and each probe generates 10 tokens. Therefore, for $k$ reasoning tokens the total number of probe tokens is $(k/128)\times 10 \approx 0.08k$, i.e., only a small fraction of the overall generation.
>
> We also evaluated CORE on the AMC23 dataset (which contains simple questions with short reasoning chains) using Deepseek‑R1‑Distill‑Qwen‑7B; the results are shown below. On short reasoning chains CORE achieves comparable accuracy and latency to vanilla decoding. This is expected: for short reasoning chains there is little redundant reasoning to remove, so the potential token savings are limited and CORE does not introduce noticeable extra reasoning latency.
>
> | Method | Accuracy | # tokens | Latency(s) |
> |----|----:|----:|----:|
> | vanilla | 89.09 | 6,233 | 199 |
> | CORE (aggressive) | 85.94 | 4,829 | 172 |
> | CORE (conservative) | 88.97 | 5,201 | 182 |
>
>
> **Q3b: More experiments on adaptive or sparse probing**
>
> A3b: Our default probing interval is 128 tokens; we also evaluated a sparser schedule using 256‑token intervals (with window sizes w=6 and w=3, respectively). Results are shown below. The 256‑token interval reduces total tokens but incurs a noticeable accuracy drop, so we use 128 as the preferred trade‑off.
>
> | Fixed interval | 128 | 256 |
> |----|----:|----:|
> | Accuracy / # tokens | 50.21 / 8,849 | 49.38 / 8,664 |
>
>
> **Q4: Analysis on lost failures (never reaching a confidence region) and confident failures (converging to a low-entropy wrong answer)**
>
> A5: We analyzed the DeepSeek-R1-Distill-Qwen-7B model on AIME25, comparing the vanilla baseline and CORE in terms of "lost" and "confident" failures and their token consumption. Results are shown below.
>
> |  | Lost Failures | Confident Failures |
> | :--- | :--- | :--- |
> | Vanilla (Acc / #tokens) | 12.5 / 21,376 | 0 / 20,717 |
> | **CORE (Acc / #tokens)** | **12.5 / 21,376** | **0 / 20,056** |
>
> **Interpretation:** For lost failures (where the model never enters a confidence region), CORE does not intervene and allows generation to run to completion, so accuracy and token counts match the vanilla baseline. For confident failures (where the model converges to a low-entropy but incorrect answer), early stopping via CORE does not change the final (incorrect) outcome but does reduce token consumption. In short, CORE reduces latency on confident failures while leaving lost failures unaffected.

---

### Official Review · Reviewer_E1ap · 2025-10-31

**Soundness:** 2
**Presentation:** 3
**Contribution:** 2
**Rating:** 4
**Confidence:** 4

**Summary:**

This paper addresses the efficiency problem of CoT reasoning in LLMs. The authors point out that while CoT enhances performance, it also leads to high computational costs and latency due to redundant token generation. The paper's main contributions are twofold: First, the authors claim to be the first to empirically reveal a global dynamic pattern of entropy during CoT reasoning. They find that the reasoning process transitions from a high-entropy Uncertainty Region to a low-entropy Confidence Region, arguing that entering the latter is a strong signal of reasoning completion. Second, based on this insight, the authors propose Confidence Region Exits (CORE), a lightweight, training-free algorithm to detect if the model has stably entered the Confidence Region, thereby triggering an early exit. Experiments show that CORE achieves a superior Pareto trade-off between token consumption and accuracy compared to baselines like DEER and Dynasor.

**Strengths:**

- Significance: The paper addresses an very important and widespread problem: the computational efficiency of CoT reasoning. Improving efficiency is critical for deploying powerful LLMs in resource-constrained environments and for reducing the cost and latency of large-scale inference, making it highly significant to the LLM community.

- Originality: The paper's strongest contribution is its pioneering empirical investigation to reveal the underlying two-phase entropy dynamics of CoT reasoning. This novel model of partitioning the process into an Uncertainty Region and a Confidence Region provides the field with a new perspective for understanding LLM internal states and reasoning redundancy.

- Quality: The experimental design, on the metric of token-accuracy trade-off, is solid. The results show that CORE achieves a Pareto-optimal trade-off over existing early exit methods on multiple models and benchmarks.

- Clarity: The paper is well-written and easy to follow. Figure 1a, in particular, is an exceptionally clear and powerful visualization that effectively communicates the authors' core insight. Figure 2 also clearly and concisely explains the CORE algorithm's mechanism.

**Weaknesses:**

1. The proposed CORE method relies on static hyperparameters ($\beta$ and $w$), which is its most significant flaw. The authors' own data proves this approach is brittle:
  - Model-Sensitive: Table 4 shows a 10x difference in $\beta$ (0.2 vs 0.02) across models, indicating a lack of portability and requiring expensive, model-specific tuning.
  - Task-Sensitive: Figure 4 shows it fails to adapt to task difficulty within the same model. On hard problems for Qwen, token savings lead to a proportional accuracy drop, a very poor trade-off, proving the threshold prematurely cuts off necessary thinking.

2. The algorithmic innovation of CORE itself is minimal. It is an incremental engineering improvement over existing confidence-based methods like DEER. DEER (single-step entropy $\le \beta$) is just a special case of CORE with $w=1$. Adding a sliding window ($w$) is a standard signal-smoothing technique (to avoid the glitch in Figure 1b, left) and does not constitute a major conceptual innovation.

3. The claim about efficiency in this paper is not rigorous. The delay analysis in Section 4.4 is misleading. To demonstrate the performance of CORE, the author should also provide a latency comparison among CORE, DEER, and Dynasor, as all three introduce an extra probing overhead that is currently unaccounted for.

**Questions:**

1. The paper's core claim is to improve efficiency, which implies reducing not just token cost but also practical latency. Your analysis in Section 4.4 compares CORE to Vanilla to claim that probing overhead is minimal. This comparison is considered to be invalid as it conflates the gains from exiting early with the costs introduced by probing. To fairly assess efficiency advantage of CORE, could you please provide a latency comparison between CORE, DEER, and Dynasor on the same configuration?​

2. Table 4 reveals a 10x order-of-magnitude difference in the $\beta$ threshold between DeepSeek-R1-Distill-Qwen-7B (0.2) and Qwen3-14B (0.02). This seems to contradict the paper's claim of seamless adaptability. Can you explain the reason for this drastic variance? If every new model requires an expensive tuning on a large dataset (Bespoke-Stratos-17k training split in your setting), how can the method be considered lightweight or seamless-adapted?

3. Figure 4 shows that on hard problems for Qwen3-14B, token savings lead to a proportional accuracy drop. This is a very poor trade-off and strongly suggests CORE cannot simultaneously adapt to both easy and hard tasks. Does this imply that one of your core contributions (dynamics of entropy and accuracy across reasoning stage) fails when strong models tackle hard problems?

4. We acknowledge your discovery of the two-phase entropy pattern is novel. But we consider CORE itself an engineering improvement (using a standard signal smoothing technique) specifically designed to solve the failure mode of DEER to premature exits on single-step entropy glitches (as shown in Figure 1b, left). How do you explain your contribution?​5.The paper mentions in Section 2 that $T_i$ is a "sequence of tokens (e.g, fixed interval of tokens)." Please clarify the value of this fixed interval used in your experiments and your justification for choosing it.

---

> ### Author Response · Authors · 2025-11-21
> **Response to reviewer E1ap**
>
> Thank you for your feedback. We are pleased that the reviewer recognizes the significance, originality, quality, and clarity of this work. We address your concerns as follows.
>
> **Q1 (Weakness 3): Latency Comparison with DEER and Dynasor**
>
> A1: We conducted the latency comparison on the AIME25 dataset using Qwen3-4B-Thinking-2507. As shown below, at a comparable accuracy level (~76.5%), CORE reduces both token consumption and latency compared to DEER and Dynasor. This confirms CORE's efficiency.
>
> | Method | Number of tokens | Latency (s) |
> | :--- | :--- | :--- |
> | Dynasor | 22,643 | 778 |
> | DEER | 21,654 | 743 |
> | **CORE** | **20,744** | **718** |
>
> **Q2 (Weakness 1): The hyperparameter $\beta$ is model-sensitive. Tuning on Bespoke-Stratos-17k dataset is expensive.**
>
> A2: **Model-Sensitive**: The difference in $\beta$ value reflects the intrinsic confidence distribution properties of different model architectures, rather than a flaw in the method.
>   - **DeepSeek-R1-Distill-Qwen-7B** is a distilled model. Such models often exhibit a smoother output distribution (higher entropy baseline) or more signal noise, thus requiring a relatively higher $\beta$ (e.g., 0.2) to reliably confirm it has entered a low-entropy state.
>   - **Qwen3-14B** is a more powerful base model. When it enters the "Confidence Region," it tends to be extremely certain (very low entropy). Therefore, a stricter $\beta$ (e.g., 0.02) is needed to capture this sharp transition.
>
> We verified this by measuring the mean entropy on 100 samples from the Bespoke-Stratos-17k dataset:
>
> | Model | DeepSeek-R1-Distill-Qwen-7B | Qwen3-4B-Thinking-2507 | Qwen3-14B |
> | :--- | :--- | :--- | :--- |
> | **Mean Entropy** | 0.52 | 0.32 | 0.21 |
> | **Optimal $\beta$** | 0.02 | 0.05 | 0.2 |
>
> Thus, the variance in $\beta$ is an adaptation to the model's intrinsic entropy dynamics.
>
> **Tuning is lightweight**: We clarify that we used only **100 random samples** from the Bespoke-Stratos dataset to determine $\beta$ from [0.01, 0.02, 0.05, 0.1, 0.2, 0.5]. This process is computationally negligible and only needs to be performed once per model, justifying
> the "lightweight" claim.
>
> **Q3 (Weakness 1): Task-sensitivity and performance on hard problems.**
>
> A3: **Conclusion:** The results confirm CORE's key advantage: it adaptively targets actual redundancy rather than indiscriminately shortening reasoning chains. Thus, CORE achieves high token savings on easy tasks while remaining conservative on hard tasks under the same $w$.
>
> **Analysis:** To understand the redundancy distribution, we analyzed the token usage of Qwen3-14B across different task difficulties. For each problem, we computed the average number of tokens required by the model under vanilla decoding, and compared it to the minimal token count when the ground truth appears in the probing answer (i.e., immediate truncation upon correct answer). Redundancy is defined as the difference, normalized by the vanilla length. Results:
> | Difficulty | Easy | Difficult |
> |:---|:---|:---|
> | Redundancy | 52% | 13% |
>
> The data shows that easy problems exhibit high redundancy (52%), which CORE effectively prunes. In contrast, difficult problems have low redundancy (13%), meaning most tokens are essential.
>
>
> **Q4 (Weakness 2): The algorithm innovation of CORE.**
>
> A4: Our primary contribution is the discovery of the **"Uncertainty Region $\to$ Confidence Region"** dynamic. CORE is the algorithmic realization of this insight: we redefine reasoning completion as a **stable state** rather than a momentary point. The sliding window ($w$) is not merely a smoothing technique but a necessary proxy for detecting this stability, solving the "premature exit" failure mode of prior point-based methods.
>
>
> **Q5: On the Probing Interval**
>
> A5: We set the probing interval to 128 tokens based on empirical trade-offs. We compared intervals of 64, 128, and 256 tokens (adjusting window sizes $W$ accordingly). As shown below, an interval of 256 degrades accuracy, while 128 achieves similar accuracy to 64 but with fewer probes. Thus, we selected 128 as the optimal balance.
>
> | Fixed interval | 64 | 128 | 256 |
> | :--- | :--- | :--- | :--- |
> | **Acc / # tokens** | 50.21 / 8,869 | 50.21 / 8,849 | 49.38 / 8,664 |

---

### Meta-Review · Area_Chair_GMRr · 2026-01-04

**Summary:**

This paper introduces Confidence Region Exits (CORE), a training-free early exit method designed to improve the efficiency of Chain-of-Thought (CoT) reasoning in LLMs. The authors observe a two-phase entropy dynamic: an initial Uncertainty Region (high entropy) followed by a Confidence Region (low entropy). CORE utilizes a sliding window to detect when a model stably enters this confidence region to halt generation. Reviewers consistently pointed out that CORE represents an incremental engineering improvement over existing methods like DEER. The primary technical contribution—adding a sliding window for signal smoothing—is a standard technique and does not constitute a significant conceptual breakthrough. While the authors correctly identified that Reviewer EDuu's citations were potentially hallucinated/irrelevant, the broader consensus among the other three reviewers (Elap, YArb, 7NGG)—who all rated the paper below the acceptance threshold—remains consistent regarding the paper's technical weaknesses.

**Reviewer Concerns:**

The authors provided a direct latency comparison between CORE, DEER, and Dynasor on the AIME25 dataset, showing that CORE achieved lower latency at similar accuracy levels. New experiments on the AMC23 dataset (short reasoning chains) demonstrated that CORE does not significantly degrade performance or add noticeable latency in these cases. Reviewers 7NGG and EDuu argued the contribution is an incremental engineering adjustment. While the authors claim a "conceptual leap" by defining reasoning completion as a stable state rather than a point, the implementation (a sliding window) remains a standard heuristic. It lacks theoretical grounding to prove why low entropy guarantees reasoning soundness.

**Reviewer Scores:**

Reviewer Elap concern that the method requires "expensive, model-specific tuning" remains functionally true. The proportional accuracy drop on hard tasks was also acknowledged but not solved by the authors. Elap likely would have remained at a 4, viewing the method as a narrow engineering fix rather than a robust framework. Reviewer 7NGG specifically asked for a "more principal tuning method" , which the authors did not provide, instead sticking to their 100-sample heuristic. This reviewer would likely have maintained their 4, citing a lack of methodological depth.

---

### Decision · Program_Chairs · 2026-01-26

Reject